# Teachers’ Perceptions of Supporting Young Carers in Schools: Identifying Support Needs and the Importance of Home–School Relationships

**DOI:** 10.3390/ijerph191710755

**Published:** 2022-08-29

**Authors:** Amy Warhurst, Sarah Bayless, Emma Maynard

**Affiliations:** 1Department of Psychology, University of Winchester, Winchester SO22 4NR, UK; 2Department of Child & Family Health, King’s College London, James Clark Maxwell Building, London SE1 8WA, UK

**Keywords:** young carers, school, identification, support

## Abstract

Recognition and support for young carers has improved steadily in the past two decades; with stronger legislation and more visibility and awareness of the challenges that many of the YC face, especially with respect to their education. Recent UK-based initiatives providing toolkits and guidance for school staff have provided much needed direction for schools, to support the loosely defined statutory requirements. The aim of the current research was to hear from school staff about their experiences in identifying and supporting young carers, to better understand any enablers and barriers. The thematic analysis of the interview data from 18 school staff was organized into two main themes: perceptions regarding the characteristics of young carers; and perceptions regarding the importance of home–school communication. Each superordinate theme contained several sub-themes. Overall, the teachers perceived many difficulties identifying young carers who did not volunteer this information and felt that the main enabler of identification was the trust relationships between the school and the pupil and parents. Once identified, the schools perceived the main areas of need that they could provide support for were the emotional wellbeing of the pupils and additional academic opportunities. They spoke too of the difficulties balancing the provision of this extra support within the constraints of the school context, both in terms of the school day, and the competing priorities relating to academic and social–emotional needs. School staff recognized that extra time outside of school was difficult for young carers to attend. Other subthemes are discussed with consideration to enablers and barriers. The implications for the dissemination of good practice, and addressing policy are considered.

## 1. Introduction

The children and young people who perform caring responsibilities to support a family member with a physical or mental disability or other health condition are referred to as young carers. It is understood that young care-giving goes beyond the usual chores and household-help that children may perform in everyday family life. In many cases, the young caregiving extends to the types of care and duties that fall within the remit of social care (e.g., administration of medicine, personal care, managing the household, looking after siblings, as well as emotional care and managing parental mental ill health; see [1]). The importance of understanding young caregiving was first highlighted in the 1990s [2,3,4]. The Carers (Recognition and Services) Act [5] offered a legal definition of young caregiving, which entitled young carers to their own needs assessment (independent of the cared-for person). Since 2014 [6,7], the guidance and legislation was consolidated and families have an explicit, holistic entitlement for needs’ assessments, with a greater protection and recognition of the needs of young caregivers [8]. The most recent census data [9] note that there are more than 177,000 young carers in England and Wales, with children often taking on this role at a very young age, (the authors of [10] report a mean age of 9 years). It is well acknowledged that the true number is likely to far exceed this, as many families may not choose to disclose or even recognize the level of caring duties carried out by young people, with families being considered “hard to reach” and young caregiving a “hidden” phenomenon [11,12]. Compounding this complexity in estimating the scale of young caregiving are the different cultural expectations around family structures and the responsibilities around family caregiving (e.g., [13]) where assuming responsibility for some care tasks is reported as being perceived as a family responsibility, rather than as providing care (e.g., [14]). The families from under-represented ethnic minorities have also reported a perceived and experienced stigma around accessing services, for example, relating to worries whether support would be culturally appropriate [15,16,17]. The young carers from some ethnic minority groups may also be less likely to self-identify as a young carer [18]. In some families, the previous experience of culturally insensitive services may reduce their faith in, and inclination to reach out for, support [17]; thus, known inequalities impact the actual and perceived support [19,20].

The definition of young carers is heterogenous given the diverse range and extent of the responsibilities that these young people take on, and the different impacts that these responsibilities have on their lives [21]. The study of how young caregiving affects children and young people addressed a range of factors, including psychological, social, and educational considerations [22,23]. Although there are some perspectives of resilience and steeling effects [24], there is also a significant potential for detrimental effects and outcomes for this group. The young carers are at risk of psychological difficulties, including poor mental health [25,26], also extending into early adulthood [27]; perceived stress [28] and emotional difficulties [29]; social challenges relating to peer relationships [30]; and potential bullying/stigmatization [10,31]. The research has recognized the emotional burden of living with a family member who has long-term illness, mental ill-health, or disability [25,32], and that this is not necessarily further exacerbated by the caring role per se [28]. Nonetheless, a growing body of research has outlined how the caregiving responsibilities may interfere with everyday life, especially with respect to education and social experiences [33].

Schools play a pivotal role in identifying and supporting the vulnerable pupils and those who need emotional support [34,35]. The schools are involved in the identification of the young people who have caring responsibilities, offering referrals, providing sources of support, and engaging families in safeguarding and pastoral care. The children who begin caring at younger ages, during primary school, are most often identified by their teachers or support staff [36]. The support that the schools offer differs widely, and may include dedicated safeguarding leads with special responsibilities for YC, lunchtime clubs, awareness raising, and counselling services (see [16]). While the schools may represent a source of positive support [37], and potential places of respite, for many of the young carers school may simultaneously be a source of additional worry and stress [36]. The surveys of young carers in schools have identified two main themes, one relating to the academic concerns held by young carers (e.g., managing school work alongside their other responsibilities), as well as stress relating to social relationships (e.g., being accepted, making, and maintaining friendships, social exclusion) (see [16,38]).

Some of the significant concerns about the young carers in schools are their higher levels of absenteeism [10], lower educational attainment [39], and the emotional toll related to the challenges of balancing caregiving with educational engagement [40]. The young carers are known to be at a high risk of social and economic disadvantage, and to experience restricted educational opportunities [39]. Particular concerns are the increased absenteeism, and the emotional and physical effects of caregiving interfering with the school day [2]. These effects are demonstrated by, on average, whole-grade differences at GCSE level between the young carers and their peers [41]. The detrimental effects on the educational experience, opportunities, and academic outcomes are explained by two main factors. One relates to the physical load of young caregiving, which leaves young carers feeling physically tired, exhausted, and often underprepared and running late for school. The second relates to the emotional load related to caregiving, with young carers carrying a burden of worry, guilt, and increased cognitive load related to the responsibility of caring [42]. This cognitive pre-occupation will also be a contributing factor to the lower ability to concentrate, and engage with school work. Stress and anxiety for example, are well known for restricting the cognitive processing capacity and thereby interfering with learning [43,44].

In recent years, there has been an increasing awareness of the need to support young carers in schools. Young carers were given a specific mention in the Ofsted Common Inspection Framework, since 2015 [45], “Young carers are a vulnerable and disadvantaged group, as a result, these pupils will have specific needs to which the school must respond”. The Carers’ Trust and The Children’s Society [46] jointly developed a set of principles, “Young carers in schools toolkit”, aimed at providing resources and guidance and sharing good practice amongst the schools in England and Wales. These initiatives have driven better awareness, recognition and support over the past decade; however, school staff continue to express the need for more support and resources [36]. Listening to the voices of the young carers themselves, it is also clear that better support for attending school and school flexibility is felt to be supportive and helpful [16,47].

The aim of the current study is to explore the teachers’ perspectives on the provision of support for young carers in primary and secondary schools in the Hampshire area, to highlight the enablers and examples of good practice, and to identify the common areas of challenge for the key stakeholders to become aware of, and to work towards reducing or overcoming these barriers.

## 2. Methodology

Participants: The interviewees were 18 teaching staff, including head teachers, SENCos, and pastoral leads in Primary and Secondary schools in Hampshire (see Table 1). Pseudonyms were used throughout, to protect the participants’ identities.

Materials: A total of 18 individual one-to-one interviews with a member of the research team. All interviews followed the same semi-structured interview schedule (see Appendix A), with the aim being to guide the participant through various aspects of identifying and supporting young carers in school and elicit their experiences.

Procedure: The research team sent out email invites to all of the schools in Hampshire (infant through secondary). These emails detailed the research area of interest, a brief outline of the research aims and procedure, and a consent form. Those who wished to participate could ask any questions, complete the form, and arrange a mutually convenient date/time for a face-to-face or online interview with a member of the research team. The interviews lasted an average of 61.7 min (SD = 10.51; range = 47–86 mins). The participants were able to ask any questions before and after their interviews and were reminded of their right to withdraw their data. The participants received a copy of the transcripts by email and were invited to make amendments, approve of, or withdraw from the study. No participants chose to withdraw, and following final approval, a thematic analysis was conducted. This followed the six stages as detailed by Braun and Clark [48]:

“*1. Familiarizing yourself with your data. 2. Generating initial codes. 3. Searching for themes. 4. Reviewing themes. 5. Defining and naming themes. 6. Producing the report*”.(p87)

Personal Reflection:

The approach used in this research study was reflexive and inductive [49], using a semi-structured interview schedule, and active listening and involvement, to draw out the individualistic experiences of teaching staff. The interviews were conducted by the members of the research team with no prior experience of young carers, either in an academic or a personal capacity. The creation of the interview schedule and the analysis of the transcripts, however, were undertaken by the researchers familiar with common themes within the research literature, and personal familiarity with the young carers, and is therefore subject to preconceptions, researcher bias, and good intentions towards supporting this group. The analysis too, is subject to researcher bias, as reflexive thematic analysis requires researcher-led meaning making [50], interpretation, and the co-creation of understanding between ourselves and our interviewees [51]. Therefore, the subsequent findings and reporting are heavily context-dependent and arise from the interconnection of the practitioners, the interviewers, and the researchers.

## 3. Results and Interpretation

To begin the analysis, the codes were initially formed per participant, and were later grouped together as themes arising from the research [48,49,50]. The entire analysis was completed manually, using post-it notes, Excel, and Padlet. These themes reflect the experiences and perceptions of the school staff who were interviewed; with some positives stated amongst the clear indications of young carer stress and distress, although this is not evidence of the young carers’ experience, but of the interviewee’s beliefs about them. Ten sub-themes emerged from the thematic analysis, which are grouped into two superordinate themes shown in Table 2.

The first superordinate theme, school perceptions regarding the characteristics of the young carers, incorporates those aspects and experiences which are unique to each individual young carer. The second superordinate theme, school perceptions regarding the importance of the school–family relationship, includes the inter-connections between the school, the young carer, and the young carer’s family.

### 3.1. Theme 1: School Perceptions Regarding the Characteristics of Young Carers

This superordinate theme contains six sub-themes, integrating school-based academic attainment, social and emotional needs, home-based factors, caring duties, and resilience/positive gain.

#### 3.1.1. Emotional Needs

Every interviewee indicated how important the emotional wellbeing of their pupils was, and how this was of primary importance, over and above that of academic attainment (see also [52]):

“*I mean, we are not an exam factory, we are about everyone trying to be equipped with what they need, to be the best possible adults they can*”;(Jan)

“*I think until a child is receiving the emotional support (they may even feel stable) that always accessing learning becomes a barrier if they have something emotionally that they’re dealing with”;*(April)

*“It means that went by the time he gets to school. Sometimes he’s just he’s overloaded and not ready to learn because he’s so overwhelmed by what’s happening at home… I think it’s just making sure that they have the time, whether that’s time to settle in the morning, time to off load, time to share, and then refocus*”.(Chris)

The quotes suggested a strong conviction for the need to address the wellbeing of young carers first and foremost, and that the efforts to promote their academic achievement were secondary to, and perhaps ineffective without, this wellbeing support [53]. These concerns covered a range of aspects, including emotional wellbeing as a foundational resource to be able to learn effectively, to the wider objectives of education, beyond exam attainment, to include transformational experiences and the growth of the individual, represented in the factors comprising the “hidden curriculum” [54].

#### 3.1.2. Caring Duties

In addition to commenting on the emotional burden of young caregiving, interviewees described how the impact of caring duties could affect the pupils throughout the school day:

“*Quite often they’re exhausted, they’re tired, they struggle to concentrate*”;(Sandy)

“*The loads that they carry is huge*”;(Katrina)

“*There’s an awful lot of thought within school of ‘I wonder what’s happening at home’, and then when they’re at home doing their caring, there may be the thought ‘Well, what are my friends doing? What are my peers doing? What am I missing out on?’ So there’s possibly an awful lot more … emotional working memory being used, so that impacts on their education, their ability to engage with education*”;(Jenny)

“*Attainment-wise children don’t make progress until their emotional needs are met … they’ve got to feel happy and happy in themselves… with the right mindset before they can do any learning whatsoever. Often, they worry about what they’ve left behind in the morning, you know, have they left their mum or their dad in a bit of a state because it’s time for school. …it’s so important, isn’t it, that gets sorted first*”.(Sophie)

The effects of the varied duties and responsibilities carried out by the young carers seemed to be both physical (e.g., tiredness, exhaustion) and mental (related to the mental burden of responsibility, as well as worry). Both of these have consequences on the ability for the children to learn effectively. The effect of fatigue on cognitive performance is well known [55]. However, mental load and worry also compromise cognitive performance, for instance, where the worries are distracting, and where anxiety or rumination may burden the cognitive-processing capacity (e.g., attention and working memory), restricting the capacity for learning [43,44]:

“*And perhaps, particularly some of the younger ones, are having… to deal with things that are kind of beyond their chronological age, and it’s just a distraction from the business of learning and focusing on their own progress in school… it depends on how serious the situation is, but I think just getting through the school day, I’m aware that some of the young carers will be thinking about siblings or thinking about parents just as they go about everyday business*”.(Hannah)

Children’s personal burdens were clearly not left behind once they arrived at school, but were a constant and continuous mental and physical challenge that interfered with the young people’s ability to engage with their learning, as well as their capacity to learn to their potential.

#### 3.1.3. Academic Needs

In their interviews, many of the interviewees commented directly on the academic needs of the young carers, noting that they were likely to have more difficulties than their peers:

“*So the support plan goes through the impacts of being a young carer, the struggle to get homework done, low attendance and lateness, feeling guilty and a heavy burden of responsibility, strain on friendships, feeling anxious and worried all the time, impact on grades and GCSE results, and restricted childhood and opportunities*”;(Cara)

“*The impact for some of these students is utterly life changing. And we’re not just talking about attainment and progress grades, we’re talking about real life for real people*”.(Harriet)

Several of the teachers spoke about their concerns regarding the provision of emotional support for the young carers and the consequential loss of classroom learning and how this affected academic needs. They were very keen to support the emotional needs of these pupils, but were aware that the majority of the support the school offered was during lesson-time, in the form of one-to-one sessions with staff. This was particularly problematic for the young carers as they may struggle to catch-up with missed work at home:

“*Whenever we have one-to-one sessions with students… we are always aware that we don’t want to keep them out of lessons for too long. Especially if they are a young carer…they’re obviously low on attendance of their lessons, we don’t want to keep them out of their lessons where they miss out on their education. So, there is that time constraint of being able to put that support in place*”;(Cara)

“*Obviously we offer the support, we offer to catch up, fill in the gaps, bits they’ve missed. But then, for me dealing with the other side of it, is that too much pressure, is that putting even more on them? I honestly I don’t know*”;(Sandy)

“*Some students don’t want to miss the lesson, which is, again, absolutely fine*”.(Anya)

Alternatively, the schools were sometimes able to provide emotional support during lunchtime, but this had its own difficulties, due to time-constraints:

“*We offer lunchtime sessions for young carers here, but our lunch time here is only 30 min*”;(Cara)

“*Students can’t always stay after school, which is why we try and do it in lunchtimes*”;(Anya)

“*It’s quite difficult. We’ve tried lunchtime, but …they’re only half an hour, so by the time they get there, it’s nearly time to go. Again, it’s got to be stacked against the time that the young carers group can meet as well…we had to sort of almost fit it in around the key stage four timetabled lessons where they weren’t having an assessment in, so maybe a PE lesson or an enrichment lesson, because the school day is rammed*”.(Jan)

Some of the schools offered emotional support after school had finished, but this too was difficult, as the young carers were often needed at home and were unable to extend the school day in this way:

“*If you are a young carer you are trying to think about school, but you are also thinking about home. And I think the barrier is sometimes those children staying after school to be able to take part in things. Even down to the kind of more academic activities like revision classes, that might be more challenging for some of these children because they feel that they have to get home, that they have got a commitment and an obligation almost to getting home and focusing on whoever they’re caring for at the expense of their own education*”;(Hannah)

“*At home it may be difficult for them to join perhaps an after-school activity or club, because they may have to get home to cook the dinner or pick up a younger sibling from school and do whatever chores are needed. So, their lives are busier*”.(Giulia)

That said, all of the teachers spoken to did acknowledge that academic work, both classroom, and home-work, was also something that the young carers may need some additional support with:

“*I think school for a young carer is harder, it’s a lot harder… With a lot of the home situations, obviously it does affect schooling, and it does affect attendance and whatnot.*”;(Sandy)

*“If you’ve got to stay in and look after a parent…it could effect things such as homework, reading, spelling, all these little things that you might expect to do at home, therefore [it] might have a knock-on effect in terms of how much progress they’re making*”;(Chloe)

“*I think that the availability from parents for schoolwork, for encouragement, for homework, those kind of things, I think it’s notable to me that isn’t quite so present for young carers.*”.(Harriet)

The schools were keen to support their young carers’ academic needs as much as possible, and many ran homework clubs or gave additional leniency with deadlines etc.:

“*If there is a young carer who hasn’t done their homework, that we have that understanding and we don’t necessarily apply the normal sanctions that we would because we recognize that kind of vulnerability that they have*”;(Jan)

“*If they’re struggling to find a space to do their homework or they’re struggling to do their homework because of other responsibilities then some leniency, perhaps should be employed, so that they don’t feel so pressurized*”;(Guilia)

“*We will offer extra time at lunch and break for them to complete homework*”.(Cara)

It became apparent from hearing the interviewees’ experiences that the academic needs of young carers can be complex, and often difficult to support within the constraints of the school day. There was a clear need for additional support, but providing this alongside the emotional support often proved difficult. The school staff seemed to need to manage a balance between prioritizing the emotional and wellbeing needs and the opportunities for learning and academic support, while not sacrificing timetabled teaching (see also [52]). The experiences of the teaching staff seemed to highlight the need for more wellbeing support both in and outside of the school setting, in order for young carers to be able to better engage academically and be able to access the academic support effectively, which requires access and referral to support from outside agencies (e.g., see [46,56]).

#### 3.1.4. Social Needs

The discussion of managing support within the school day indicated an additional theme about the social–emotional challenges that young carers may face. The teachers highlighted that the school provides a place for meeting the social needs of pupils, but that this was not always straightforward for the young carers. Previously mentioned issues, such as higher stress levels, lower attendance rates, and difficulties attending after-school clubs, contribute to this, and also the lack of opportunities to meet friends after school and at weekends. Being a young carer may result in feelings of isolation or struggling to maintain and manage friendships:

“*I also think because of their commitment to caring they may not have such a full and active social life. The activities outside of school that are important for somebody’s personal development*”;(Hannah)

*“If your ability to socialize, meet up with your mates, going to parties or just to hang out at friends’ houses is limited or can’t always happen because parents can’t take you there, or you can’t take yourself [because] you’ve got responsibilities…it might affect those friendships in school. So, it might make those friendships a bit difficult*”;(Chloe)

“*They seem to miss out at each opportunity, so if there is an after-school event, or if there is a different sports events or whatever, it can be logistically really difficult*”.(Katrina)

This is particularly concerning as the young carers would benefit from having peers to talk to about their worries [30], and the teachers recognized the need for them to have a peer group and shared experience:

“*I think the fact that they knew that there were others of their peer group [that were young carers], that it wasn’t just an adult saying, we respect what you’re doing and we notice what you’re doing, it was the fact that there was a group of them that could then have those conversations and then have the opportunity to talk and to offload themselves as a group*”.(Sophie)

Teachers also recognized that school itself could be a protective aspect for Young Carers, providing supportive adults, structured and predictable hours in the day, and the chance to socialize with other children and young people:

“*I think to young carers, school is their time, where they play, they don’t have to think about looking after anybody else because they’re looked after and the roles in schools are very clearly marked…so they can relax and know their place as a child. Whereas actually outside of school, they’ve got a lot of different roles that they take on board*”;(Katrina)

“*And [school should be a place to] talk about your hopes and dreams for the future. So, there’s similarity there [with their peers] for sure, they’re still children, who enjoy playing and laughter and humour in all those different ways*”.(Katrina)

This theme highlights the complexity of the difficulties that young carers face, and the extent to which their experience of the school community, and the social opportunities it offers, is restricted for this group of children. This seems to have a dual impact, both restricting the opportunities for fostering friendships and a peer group, but also limiting the social support that the friendships and peers can offer, which is especially important for these young people who are often carrying an emotional burden far greater than might be expected for their age. It represents a clear challenge for fostering appropriate opportunities for young carers to be able to socialize, especially with others in similar situations to themselves. The schemes that have proven very successful include young carers projects [57], and in-school support groups [58].

#### 3.1.5. Home/Family Circumstances

Each participant commented on the individuality of their young carers and were keen to stress that this meant that they all had different experiences and needs, and that these would fluctuate depending on how their home or family circumstances changed:

“*We’ve got about 20 Young Carers within school at the minute who do an actual caring role for their parents where they help them out of bed and help them with daily things. And then I would say the rest of them are children whose parents might have mental health issues, or where they have a sibling who has needs*”;(Chiara)

“*Some of them have different, I suppose, levels of responsibility, from having a parent with mental health issues to having a parent who’s a quadriplegic and some other health issues in addition to that*”;(Guilia)

“*I know a number at the moment who are kids of terminal cancer patients. We’ve certainly got a handful of those who are really aware of, and they’re on the radar at the moment*”;(Harriett)

“*Some are children who maybe a parent has a mental illness, or substance misuse*”.(Mia)

They highlighted the schools’ flexibility in how they could help these students, taking a bespoke approach, as far as possible, to each pupil:

“*The action plans are created separately because the circumstances around the individual families are vastly different… Sometimes the care becomes much more intense, and sometimes the care doesn’t need to be so intense. So at times when the child is, you know, fully involved it, You know, we have to be on it, straight, all the time, when other times it settles down and then we can step back a little bit*”;(Sophie)

“*Once you understand and you know what is going on with that family, you then action something to support that child or that family. But if you don’t know the details of it, then you can make assumptions that aren’t true, and that are actually inaccurate and that’s really unhelpful*”;(Katrina)

“*I think they face all of the challenges that teenagers growing up in 2021 face, but they’ve got kind of added levels based on their home circumstances which other children might not have. At the end of the day, they are young people growing up who share a lot of the challenges but there may not be the support or the resources to help them to meet those challenges because of the caring situation that they’re in*”.(Hannah)

It is clear from the responses that a one-size-fits-all approach to supporting the young carers is not necessarily pragmatic or helpful. It may provide a guiding support plan, but a great deal of flexibility is required, both for identifying these young people and understanding their individual needs. There is a need to build on the existing approaches of identifying and supporting young carers in schools, to improve knowledge, understanding and share good practice. The school staff addressed the perceptions and experiences around practical and resource limitations in their responses, which we explore and interpret in our companion paper.

#### 3.1.6. Resilience/Positive Gain

The sixth and final subtheme of perceptions of young carer-based factors is the resilience and positive gain that YCs may develop due to their situations and obligations, despite being presented with so many challenges that are mentioned.

The increased maturity and ability to be self-sufficient is something that young carers seem to demonstrate more than their peers, as a result of their extra responsibilities at home:

“*I think a lot of them [YC] seem to be a lot more mature than their peers*”;(Cara)

“*And I’ve noticed that a lot of our young carers seem to be very mature for their age*”;(Giulia)

“*and they [the young carers] were pretty self-sufficient*”.(Chloe)

Developing such characteristics is an advantage, especially for young carers, as this can lead to a decreased level of difficulties and increased mental health and wellbeing [24]. This also leads to other beneficial qualities, such as improved organizational or coping skills. Increased resilience has also been observed [24]. These are all key transferable skills which are highly regarded by future employers [59]:

“*Their organization and coping skills are not adversely affected. In fact, in some cases are stronger. Similarly, their ability to empathise and show thoughtfulness I think are not diminished, and again, are often higher because of the needs they have…I guess in future life as well, it will make them more resilient and more able to cope with situations far more than a child who hasn’t been there, a young carer*”.(Sophie)

It should be noted that despite a fair proportion of empirical research addressing positive gain and resilience, the theme was not as prominent as others identified in our analyses. While the respondents acknowledged some level of positive gain for these young people, this was largely outweighed by the recognition that much of the emotional and physical burden is not age appropriate, and may hinder healthy emotional development, despite some potential for resilience [40,60].

### 3.2. Theme 2: School Perceptions Regarding Importance of the School-Family Relationship

The second superordinate theme “School perceptions regarding the importance of the school–family relationship’ is made up of four sub-themes, highlighting the importance of communication and the openness of both the young carers themselves and their parents, parental permission, and the value of trust.

#### 3.2.1. Importance of Communication

Several interviewees mentioned the significance of communication between the families of young carers and their schoolteachers:

“*So if when talking to a student it’s clear that they are a young carer, then it’s about speaking with their parents to establish with the parent their feelings on it…We’ll have a discussion with the pupil, then make a call home to parents to discuss and whether they would like the referral to go through and if they want the support*”;(Chiara)

“*We try to involve the parents at sort of every step. So once we get a name, I would be phoning parents to discuss the difficulties within the home from a supportive standpoint…For example, “your son has told us that maybe you or another sibling is unwell and they need to do this, this and this, we have a young carers group, we can support them through” and starting those conversations*”.(Amina)

For this to work well, there needs to be mutual trust and an openness to their conversations and information sharing, which for some families is very difficult and can create a barrier for seeking help and support for their child(ren) [61]:

“*So I think for some parents, that conversation has to be really sensitively handled and the only downside is that sometimes parents feel very sad that one of their children is being affected in that way*”;(Katrina)

“*Sharing of information is the barrier. You know, um, I think sometimes with some families, um be concerned that they would be embarrassed, or you know feel that we may judge them for the fact that they are going through a difficult situation at home. I think therefore presents a difficulty with them sharing the information in the first place.*”.(Chris)

Trust, however, works both ways, and some school staff can also find this difficult:

“*I suppose there is a barrier around trust a little bit, in that how do you definitely know that somebody is a young carer if we’re not making lots of contact with parents, there is that question mark to say we’re not entirely sure. There can be a bit of a difference between what students want to say and what parents want their children to say, or not say. I think there does seem to be a fair amount of sensitivity around some of these issues, especially I suppose around mental health more recently. It seems complicated to really get proof. Maybe that’s a barrier: How do I know for sure, even if a parent contacts me and says I’ve got mental health issues, without a doctor’s note or something you know we have to work on trust, I just don’t have the time or the resource to really make that one work [to check if child is a YC]*”.(Harriet)

Such communication can help the teachers identify who is a young carer and to what extent, by better understanding their home situation and caring roles. This open communication between the parents and schools can also provide staff with more information and knowledge on young carers and how best to help them progress [62].

“*Other good things, [about identifying YC] I think it brings families and school closer together, because actually the challenges mean you know and share an awful lot between home and school which you may not in another situation*”;(Katrina)

“*I will discuss with tutors, progress leader and then usually communication is made home and then you talk to the parent and find out what the situation is going on…But communication is kept open, they know the student services email, they know they can contact us if there’s any questions or if there’s anything they think we need to do or could do to help the student. It’s same as with the open-door policy with the students, we are here for the parents*”.(Sandy)

This also allows parents to feel more involved in their child’s educational life, and to feel more comfortable discussing the difficult topics needed to help the young carer have a better relationship with school staff and have a more productive engagement with their education. This parental involvement with their child’s education is positively correlated with higher levels of pupil achievement (see e.g., [63]):

“*I think it’s all about the trusting relationship, and I think sometimes that takes time. But in general, in my experience, the more you put in, the more you get out. So, if we are regularly supporting and having open, supportive conversations that are non-judgmental and we’re ultra-clear about that, then that helps the relationship and that gets more productive involvement because they are more happier to share. I think there are times, as well, where you go to do something above and beyond or you do something a little bit extra or unexpected, that can really help*”;(Chloe)

“*And then good communication with parents. There was a census done…they estimated that there was 1 in 12 children would be young carers…but it doesn’t tell the whole story because a lot of parents won’t put down on that form that they have the disability they have, or that their child does do extra care responsibility. So yeah, good communication with parents and helping them understand that because we as a school consider their child to be a young carer, it doesn’t mean we are reflecting badly on their parenting, they’re not connected*”;(Amina)

“*Another issue is parental understanding. I understand some parents worry about what people might think or how it will look, or other people knowing their business…But I think that’s where we are. You know, picking up the phone and making those phone calls and trying to ease any worries that they might have*”.(Anya)

Establishing communication is clearly presented as a barrier to supporting young carers on several levels. Without good communication, the initial identification of children in need of support is affected, as is any subsequent support for young people, who due to their age, will need to rely on parental permission to participate in many of the support offers available. An important additional consideration is the ethnic and cultural sensitivities around the identification of YC and the appropriateness of the support services offered. The wider context of this issue is addressed in our companion paper. The final two subthemes unpack the important facilitators of communication.

#### 3.2.2. Openness

With communication making parents and YC feel more accepted within the academic environments, an element of openness from YC and their families to their teachers will most likely follow, leading to the second sub-theme. The communication between the school and parents could be vital, depending on if the young carer themselves are open about the fact of being a young carer, and inform the school themselves or not. Appropriate action, if necessary, can take place in a more efficient way if there is clear transparency from the students, so the school staff are aware of the situations and can provide more support:

“*A plan would come about, I suppose, well, either somebody has arrived that’s been very open that they’re a Young Carer, or you’ve got suspicions that perhaps somebody is being a Young Carer and you want to investigate*”;(Chloe)

“*[for pastoral care] When I first start talking to a student, I always get them to do a sheet, I think the title is something along the lines of ‘A little bit about me’, and I always ask them to fill in information about their family and home life and siblings etc. So, I try and get a lot of information from them and then often from that once they start talking to me about the home life, they then identify to me as a young carer*”;(Giulia)

“*We have in the past got to year 11 and found out about kids that have been young carers for years and we never knew about because the family hold it very tightly and the kids don’t tell us*”.(Amina)

The interviewees mentioned that the levels of acceptance from the parents vary from grateful and appreciative of support, to not engaging and disinterested:

“*For example, we picked up issues with not having enough food, and you phone up and you say, ‘we just picked this up in school, do you need to have any support?’ Some people be like ‘oh actually yes, we are in this situation that would be lovely…’ and they’re happy for you to signpost them elsewhere, or send them out the free school meal forms… And some people are like super defensive, and they get affronted and then there’s a kind of sense of, you know, keep your nose out or sense of shame about the situation…It’s very rarely a neutral response in those situations, either kind of like ‘oh thank goodness somebody’s actually finally noticed somehow’ or like ‘who do you think you are… Stop’, you know, and you have to kind of walk quite carefully and think about how you phrase things*”;(Chloe)

“*Some parents are a bit disengaged with it, and others are really grateful*”.(Maria)

It is clear that the schools need a framework guide to establish effective communication, and some described the strategies that they have developed over time to facilitate this. The authors of [24] also demonstrate the importance of “social connectedness” for young carers and having support from others, which is a result of honesty and communication between the parents, the students, and the school. The collating and sharing of best practice and tried and tested strategies may prove useful to the schools who are working on improving the support offered to young carers.

#### 3.2.3. Parental Permissions

As mentioned earlier, a key element to providing school-based support does largely rely on parental consent, which provides a further reason for the importance of parental openness and willingness to accept support and admit to being a young carer family. Almost all of the participants mentioned that without permissions, there is very limited support that the schools can provide to the children, both in-school or outside support:

“*As soon as I’m made aware that someone is a young carer, and I’ve established that, I make contact with home to say that their son or daughter has been identified as a young carer at school, and send out the parental consent form*”;(Cara)

“*We’ve got access to lots of outside agencies if we needed it, but it depends on whether parents would want that support to be put in place.*”;(Mia)

“*We’ve had a couple of occasions where that’s happened and families wouldn’t consent, we can only provide support in school, with ELSA or they’re happy with pastoral support in school, but they wouldn’t consent to a referral to [young carers support organisation] for example.*”.(Chris)

It is evident that establishing contact and communication between the school and the families is vital to providing many of the support options available to young carers. Many of the respondents’ comments reflected the barrier to providing support when it is not possible to gain parental permission. The ways in which this can be developed and fostered is exemplified by the final sub-theme.

#### 3.2.4. Trust

All of the staff members spoken to mentioned how important trust is. This was mainly focused on the trust of the young carer in a member of staff, knowing that someone in school genuinely cares and is available for them to talk to safely, and how building this relationship may require staff to make commitments over and above what is expected within their role:

“*Because obviously it’s important they feel safe and secure…if they just know that people care about them, and they’re interested in them, and they are there to allow them to talk, then that’s going to make the biggest impact on them, so they don’t feel that they are alone, and so that their mental health is therefore, you know, protected*“;(Chloe)

“*Even if it’s tiny, I think I’ve learned that young carers often just need somebody to give them a little nudge and say “Hey how you doing?”… “I understand, how are you getting on?” Even if it’s only something like that, I think that alone is just so important*”;(Harriet)

“*I think it comes down to having those really important, consistent relationships where the child realizes that you as the adult actually care, you’re not just passing the time of day. You actually care, you want to know and you want to support, and I think that’s what’s really beneficial with any child is those relationships*”;(Jenny)

“*But he [teacher] makes a phone call earlier in the morning, like before most of us are even at school, to alert that mum that it’s time to get up and the child needs to come into school. And I just think that’s amazing, that’s everyday a commitment that at that time he will make phone call to get that child up, you know, I just think that’s just amazing, because that’s added to every end of his day (.) it’s half seven every morning*”;(Sophie)

“*It’s turned into a really deep reflective conversation at half seven in the morning. So we’re talking about the fact that she [mum] can feel judged on the playground, and then she feels people have a perception of her and what can she do to change that, and is it right or wrong…from something which was initially kind of quite a tough check on what you’re doing, it’s turned into something where that relationship now is brilliant…Mum has told me she would be happy to share anything with us as a school and that she thinks the support we offer has been really helpful and fantastic*”.(Sophie)

The school staff experiences highlight the commitment required to build trust and good communication with families, and how, in some cases, this may require a commitment that goes over and above their role expectations. The importance of this trust in school staff and a sense of feeling supported, to enable pupils to thrive at school, is well documented within the research literature [64,65,66]. It is also a theme that is clearly indicated in the Young Carers report, from the perspectives of young carers themselves [16].

## 4. Conclusions

All of the respondents were actively involved in supporting YC and had longer term experience of supporting this group of young people. There were common themes about how the YC are being supported, what pragmatic systems are in place, what works well, and where the main challenges lie. While most of the respondents’ schools had a system in place whereby YC are recognized within the safeguarding procedures, it seems to be the practical day-to-day approach to supporting YC that varies, and on which the sharing of best practice may hold the greatest benefit. Indeed, recent surveys have indicated that school staff and leaders feel that more support and training would be helpful for them to better support YC [36].

Resources, such as the Young Carers in Schools toolkit and award scheme [67] and similar guides to support schools in identifying and supporting the young carers [68], are indispensable as they offer a practical step-by-step framework for supporting YC and managing referrals to outside support agencies. However, the “softer skills” of establishing good communication, trust, and relationships with the affected families are not as easily outlined or defined. It is clear that the unique and diverse family situations, caring responsibilities, and support needs experienced by young carers necessitate an individualized approach (Theme 2: School perceptions regarding the importance of the school–family relationship). It is also apparent from the narratives in our interviews that the social needs in school are complex, and that approaches that include awareness raising and providing focused peer support are effective both in identifying YC and creating a community of support and trust. The school staff expressed the perception that establishing trust and open communication with young people and their families enabled and facilitated the support processes (offering in-school support, flexibility for learning, referrals to outside agencies). The ways in which this trust and communication is established seems to be less well documented, and may often depend on the dedication and commitments of individuals. It also seems that the means by which these relationships are formed often involve staff going over and beyond their contracted duties, as well as their work-loaded time. This indicates the perception among our participants that such a level of commitment is necessary to effectively support YC, but we note this reflects wider issues of risk and safeguarding responsibilities imbibed to schools, following the Children Act 2004 [69]. Ideally, this largely hidden work should be better recognized, resourced, and best practice shared. A starting point may be that the resources about engaging “hard to reach” families could usefully be included in toolkits and resource packs [70]. The wider issues about resources and implications for policy are addressed in a forthcoming publication. The future research may need to focus on the voices of the families, parents, and young carers to better understand what is helpful in building communication, trust, and openness, to outline the experiences that were transformational in getting better support, and the characteristics of those who successfully provided support or a helping hand/listening ear. Undoubtedly, future research will need to focus on the young carers voices in this discussion.

Another key challenge seems to be balancing the emotional and psychological wellbeing against supporting academic needs. The narratives identified the need to address emotional wellbeing before being able to consider academic performance and outcomes (Theme 1: School perceptions regarding the characteristics of young carers), that the foundation for improving academic outcomes and fulfilling YC academic potential is in providing holistic support and flexibility for learning (such as making arrangements for homework, study time, time absent from school). Typically, the wellbeing of the children and young people is subjectively interpreted [71], and it is teachers’ interpretation about their pupils’ needs that we report on here, based on their professional knowledge. Therefore, we do not present our participants’ reflections as facts about the impact of being a young carer, but rather, offer the findings as a window into how school staff process their observations of young carers and realign their practices through their own lived experiences. Because their emotional wellbeing is so closely tied to academic outcomes, it is crucial to recognize the need to support young carers in reducing the physical and emotional burden they carry, to enable them to access the learning and social opportunities in a more equitable way. This will rely on further acknowledgement and policy shifts to recognize and provide support for families, in order to reduce the load on young carers. In our interviews, the school staffs’ experiences clearly described that young carers needed better access to emotional support to address the academic gap and disadvantages that are common amongst this group of young people, and that this need should ideally be met with a holistic approach to support.

## Figures and Tables

**Table 1 ijerph-19-10755-t001:** Table showing the job role and school type of participants.

Job Role	Primary	Secondary
Head Teacher	3 (Mia; Katrina; Sophie)	0
SENCo	2 (Jenny; Chris)	2 (Cara; Anya)
ELSA	1 (Chris)	1 (Francis)
Student Services/Pastoral	0	4 (Giulia; Hannah; Maria; Sandy)
Inclusion/YC Lead	2 (Chiara; April)	3 (Harriet; Amina; Jan)

**Table 2 ijerph-19-10755-t002:** Table of Themes.

School Perceptions Regarding the Characteristics of Young Carers	School Perceptions Regarding the Importance of the School–Family Relationship
Home/family circumstances	Importance of communication
Caring Duties	Openness (of parents/young carers)
Academic Needs	Parental Permission
Emotional Needs	Trust
Social Needs	
Resilience/positive gain	

## Data Availability

If you wish to access the full, anonymized, transcripts please email the corresponding author.

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
