# Peer review of "Teachers’ Perceptions of Supporting Young Carers in Schools: Identifying Support Needs and the Importance of Home–School Relationships"

_ijerph, 2022, doi:10.3390/ijerph191710755_

Round 1

Reviewer 1 Report

The key findings format needs improvements. 

Quoted verbatim responses can be trimmed down to 3-4 per theme, and analysis must be emphasised instead. 

Reviewer 2 Report

I begin by congratulating you on your work.

Supporting and helping young carers in schools is really important. In fact, these young people carry with them responsibilities and tasks that are praiseworthy. If being young in 2022 is complicated, for these YC the challenges and family contexts are even more relevant.

The suggested corrections follow.

The first sentence of the intro contains an extra “1” at the end “Young Carers1” (l33).

I think that chapter 2 could labeled as “Methodology” instead of “Materials and Methods”. (l120).

How do you conduct thematic analysis? Did you use any software (as NVivo) for analysis, or did you conduct a manual analysis?

At line 201, for “caring duties”, an “i” is missing in beginning: “n addition to”.

Reformulate the first sentence of “academic needs” (l246).

And at the appendix, two sentences are in bold without reason (for me): Does your school have a policy or action plan for young carers? AND Does anyone have responsibility for supporting young carers in your school?
